

# Effect of Secondary Ice Production Processes on the Simulation of ice pellets using the Predicted Particle Properties microphysics scheme

Mathieu Lachapelle[1,2], Mélissa Cholette[3], Julie M. Thériault[1]

[1]Centre pour l'étude et la simulation du climat à l'échelle régionale (ESCER), Department of Earth and Atmospheric Sciences,
Université du Québec à Montréal, Montréal, Québec, Canada
[2]Flight Research Laboratory, National Research Council Canada, Ottawa, Ontario, Canada
[3]Meteorological Research Division, Environment and Climate Change Canada, Dorval, Quebec, Canada

*Correspondence to*: Mathieu Lachapelle (lachapelle.mathieu@courrier.uqam.ca)

**Abstract.** Ice pellets can form when supercooled raindrops collide with small ice particles that can be generated through secondary ice production processes. The use of atmospheric models that neglect these collisions can lead to an overestimation of freezing rain. The objective of this study is therefore to understand the impacts of collisional freezing and secondary ice production on simulations of ice pellets and freezing rain. We studied the properties of precipitation simulated with the microphysics scheme Predicted Particle Properties (P3) for two distinct secondary ice production processes. Possible

improvements to the representation of ice pellets and ice crystals in P3 were analyzed by simulating an ice pellet storm that occurred over eastern Canada in January 2020. Those simulations showed that adding secondary ice production processes increased the accumulation of ice pellets but led to unrealistic size distributions of precipitation particles. Realistic size distributions of ice pellets were obtained by modifying the collection of rain by small ice particles and the merging criteria of ice categories in P3.

**1 Introduction**

Freezing rain is among the most hazardous weather phenomena in North America. In January 1998, freezing rain events caused 35 deaths and more than 4 billion Canadian dollars of damage in southeastern Canada (Public Safety Canada, 2013). More recently, in April 2023, 1 million people lost power after the accumulation of more than 30 mm of freezing rain in southern Quebec (Duchesne, 2023; Hydro-Québec, 2023). In addition to its impact on infrastructure, freezing rain causes more than 10

aircraft accidents every year by accumulating over critical aircraft components (Green, 2015). The occurrence of freezing rain is also associated with car accidents that can lead to casualties (Tobin et al., 2021). The regions that are affected by this hazardous precipitation type are expected to change due to climate change. For example, freezing rain events are expected to increase in northern Canada and decrease in southern Canada and in most of the United States (McCray et al., 2022).





Forecasting the occurrence freezing rain and climatological changes in the context of global warming relies on accurate representation of precipitation in atmospheric models. Freezing rain is difficult to represent in models because it forms in conditions that can lead to other precipitation types, such as snow and ice pellets. For example, ice pellets and freezing rain both occur when a melting layer at temperatures > 0°C develops above a subfreezing layer with temperatures < 0°C near the surface (Brooks, 1920). In these atmospheric conditions, solid particles melt in the melting layer and reach the subfreezing

layer. If the particle is partially melted, the ice that remains in that mixed-phase particle can refreeze into an ice pellet before it reaches the surface. If the particle is completely melted, freezing rain is often observed (Zerr, 1997). However, some heterogeneous freezing processes, including collisional freezing with ice crystals or immersion and contact freezing with primary ice nuclei, can also produce ice pellets (Hogan, 1985; Stewart, 1991; Stewart and Crawford, 1995; Lachapelle and Thériault, 2022). Current observational records suggest that the concentration of primary ice nucleation particles is too low to

explain the formation of ice pellets from completely supercooled liquid raindrops at temperatures > -10°C (Kanji et al., 2017; Petters and Wright, 2015; Lachapelle and Thériault, 2022).

Secondary ice production (SIP) processes can generate small ice crystals in concentrations that exceed those of primary ice nucleation at relatively warm temperatures of > -10°C (e.g. Field et al., 2017; Korolev et al., 2020; Korolev and Leisner, 2020).

It has been hypothesized that SIP processes were responsible for the production of ice pellets in some observed events (Stewart and Crawford, 1995; Kumjian et al., 2020; Lachapelle and Thériault, 2022). Among the different documented SIP processes, the Hallett–Mossop process (HM; Mossop, 1970; Hallett and Mossop, 1974) and the fragmentation of freezing drops (FFD; Kleinheins et al., 2021) are suggested to be active during ice pellet formation (Lachapelle and Thériault 2022). HM is the most studied SIP process (Field et al., 2017). Although some doubts remain concerning its physics (Seidel et al., 2023), it is thought

to occur when a fast-falling solid particle collects small cloud droplets at temperatures between -8°C and -3°C. FFD occurs through the deformation of relatively large liquid drops (e.g. breakup, cracks, bulges) during freezing (Lauber et al., 2018; Keinert et al., 2020; Korolev and Leisner, 2020). Although FFD has been observed in laboratory experiments at T > -5°C, this process seems to produce more secondary ice particles at T ~ -15°C (e.g. Keinert et al., 2020; Phillips et al. 2018). In general, the efficiency of FFD and HM processes remains difficult to establish and requires more research in the field and in the lab

(Korolev et al., 2020; Korolev and Leisner, 2020; Lawson et al., 2023; Seidel et al., 2023).

Nonetheless, the impacts of SIP can be studied using microphysical parameterization schemes coupled to atmospheric models to predict cloud and precipitation properties. So far, studies on SIP have been conducted by simulating different types of clouds and weather, including orographic mixed-phase clouds (e.g. Dedekind et al., 2023; Georgakaki et al., 2022), polar mixed-phase

clouds (Sotiropoulou et al., 2020; Zhao et al., 2021), cold frontal rainbands (Sullivan et al., 2018), cold marine boundary layer clouds (Karalis et al., 2022), stratiform clouds (Zhao and Liu, 2022), and convective weather. Convective weather represents most of this research (Field et al., 2017). In convective weather simulations, the impact of the HM process is recognized as the most important SIP process. Recent work from Qu et al. (2022) also showed that FFD could have an important impact on the



high concentration of ice particles in mesoscale convective systems. To our knowledge, simulations have never been used to
investigate and compare the impacts of HM and FFD on the distributions of ice pellets and freezing rain at the surface.

A mixed precipitation storm that included a long duration ice pellet episode (> 10 h) was recently documented by Lachapelle
and Thériault (2022; LT22 hereafter). The ice pellet episode was characterized by a warm and deep melting layer, and the
freezing of completely melted hydrometeors. A 17 mm water equivalent accumulation of ice pellets was measured at the
UQAM-PK weather station in downtown Montreal (Fig. 1). The observation of ice crystals at the surface during the entire ice
pellet episode led LT22 to suggest that the ice pellets may have formed by collisional freezing. Furthermore, SIP processes
were thought to have favored the formation of ice crystals below the melting layer. The presence of riming on many ice pellets
indicated that the HM process may have been active within the subfreezing layer. In addition, the many deformed ice pellets,
including fractioned, and bulged particles (representing 18 % and 25 % of all ice pellet particles, respectively; Lachapelle et
al., 2024), indicated that the FFD also contributed to SIP. Finally, LT22 suggested that the heavy northeasterly wind in the
subfreezing layer increased the area affected by ice pellets by transporting the slowly falling ice crystals below the melting
layer (see conceptual model Fig. 13 in LT22).

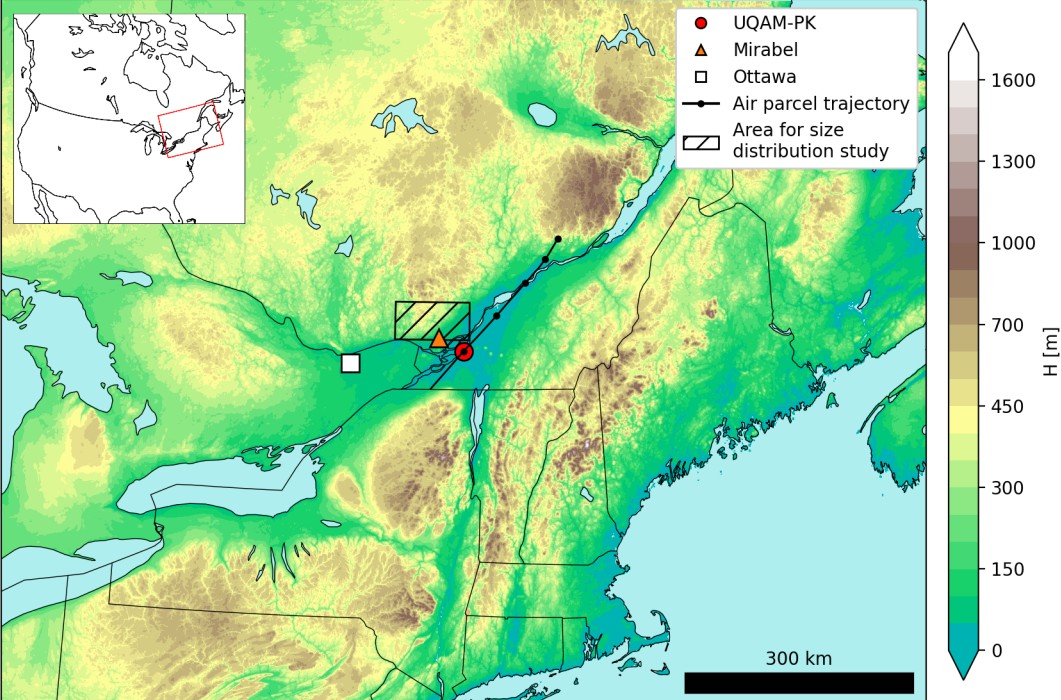

**Figure 1: Simulated domain and elevation (m). The red rectangle in the upper-left panel shows the location of the domain in North**
**America. Locations of the UQAM-PK weather station, Mirabel International Airport, and Ottawa International Airport are**
**indicated by the red circle, orange triangle, and white square, respectively. The segmented black line is an parcel trajectory of an**
**air parcel ending 500 m above UQAM-PK station at 0800 UTC 12 January 2020. The black dots along the line are located at every**



**hour along the trajectory. Finally, the hatched black square shows the area where the size distribution of simulated precipitation was studied in Sect. 4.3.**

Given that the hypothesis presented by LT22 has not been studied using simulations, the objective is to investigate the impacts of SIP processes on the simulation of ice pellets properties. We used the Predicted Particles Properties (P3; Morrison and Milbrandt, 2015) bulk microphysics scheme and high-resolution simulations with the Canadian Global Environmental Multiscale Model (GEM; Côté et al., 1998; Girard et al., 2014) of the January 2020 ice pellet storm, for which detailed observations of precipitation particles are available.


This article is organized as follows. Section 2 describes the P3 scheme, the changes made to P3, and lists the simulations used in this study. Section 3 presents the configuration of GEM model simulations. The simulations of the January 2020 ice pellet storm along with an analysis focused on the simulated particle properties are presented in Sect. 4. Conclusions are presented in Sect. 5.

## 2 Description of the P3 microphysics scheme

### 2.1 The baseline version of P3

The P3 scheme is used operationally in the GEM-based High Resolution Deterministic Prediction System (HRDPS; McTaggart-Cowan et al., 2019) from Environment and Climate Change Canada. We used a double-moment P3 for cloud droplets, raindrops, and ice. Triple-moment ice is also available (Milbrandt et al., 2021; Cholette et al., 2023), but double-100 moment ice is used herein. The two liquid-phase species have two prognostic variables, mass and number mixing ratio, and the ice-phase species have five prognostic variables per ice category in this study (Morrison and Milbrandt, 2015; Cholette et al., 2019). These are the total mass, the liquid on ice mass, the rime mass, the rime volume, and the total number mixing ratios. Ice particles in P3 grow freely from ice nucleation and vapor deposition to partial and complete riming. All simulations in this study predict the liquid on ice mass of mixed-phase particles (Cholette et al., 2019) that allows the refreezing of partially 105 melted ice particles which is a necessary process to simulate ice pellets.

In P3, ice particles can be formed/initiated via the following processes: homogeneous freezing, condensation freezing, immersion freezing with primary ice nuclei, and collection of rain and cloud by ice. Homogeneous freezing causes all cloud and rain particles to freeze when the temperature (T) is < -40°C. The resulting particles have a rime mass equal to their total 110 mass, with a rime density of bulk ice 900 kg m$^{-3}$. Condensation freezing/deposition ice nucleation occurs at T < -15°C with a degree of ice supersaturation > 5% (Cooper, 1986). P3 assumes that initial ice crystal radii are 1 μm with a rime density of 900 kg m$^{-3}$. Immersion freezing of cloud droplets and rain can occur when T < -4°C, following the volume and temperature-dependent formulation presented in Bigg (1953) with parameters from Barklie and Gokhale (1959). In Appendix A, we show that this parametrization of immersion freezing leads to the freezing of a small fraction of raindrops in the atmospheric





conditions observed during ice pellet events. Finally, cloud and rain collected by ice instantaneously freezes when T < 0°C, except for a fraction that is shed, following Musil (1970). It is worth noting that the main conclusions of this work remained the same when the simulations were run with immersion freezing turned off.

The precipitation types in P3 are diagnosed using the properties of the species located at the lowest model level. The diagnosed

precipitation types are ice crystals, snow, graupel, unmelted snow, hail, ice pellets, drizzle, and rain. To simplify how precipitation types are presented in this study, ice crystals, snow, graupels, and unmelted snow have been grouped together under the same category that we call snow. Hail and ice pellets have been grouped together under the same category that we call ice pellets. Rain and drizzle have been grouped together under the same category that we call rain. Ice pellets are diagnosed when the mass-weighted mean density of ice particles ($\rho_i$) is > 700 kg m⁻³ and snow is diagnosed when $\rho_i$ < 700 kg m⁻³.

Freezing rain and rain are diagnosed when precipitation is liquid, and the lowest model level temperature is < 0°C and ≥ 0°C, respectively. Since the model has two liquid species and many possible ice categories, a mixture of distinct precipitation phases can be simulated at the same location.

**2.2 The multiple "free" ice categories and SIP parameterization**

Using more than one ice category (nCat > 1) is possible but optional in P3 (Milbrandt and Morrison, 2016). The use of two or

more ice categories facilitates the parameterization of SIP processes since it allows ice particles with different properties to coexist. The baseline version for P3 includes only the HM process. In this study, HM will be compared to another SIP process, the fragmentation of freezing drops (FFD) adapted from Lawson et al. (2015; L15 hereafter). Observations collected in the field suggest that FFD and HM were active SIP processes during the 12 January 2020 ice pellet episode (LT22; Lachapelle et al., 2024). The two SIP processes were analyzed individually; no simulation used more than one SIP process simultaneously

because this would result in the production of secondary ice through two distinct processes freezing the same raindrop and would require the default implementation of HM to be modified.

The HM SIP process produces 350 ice splinters that are 10 μm in diameter when large ice particles (mean mass-weighted diameter > 4000 μm) collect 1 mg of rain at temperatures between -3 and -8°C. The number of splinters produced decreases

linearly as temperatures deviate from the optimal temperature of -5°C.

The FFD process is added to P3 as described in Qu et al. (2022) using L15 parametrization. L15 suggests that the number of ice crystals produced by the freezing of a drop with diameter $D$, expressed in μm, follows:

$$N_{f,L15}(D) = 2.5 \times 10^{-11} D^4 \tag{1}$$

This parametrization does not depend on the particle temperature. However, observations reported by Phillips et al. (2018), Korolev et al. (2020), and our own observations of ice pellets (Lachapelle et al. 2023) suggest that less FFD occurs when the



freezing temperature is > -5°C. For this reason, we modified the L15 parametrization by linearly decreasing $N_{f,L15}(D)$ (Eq. 1) between -6°C and -3°C and setting $N_{f,L15} = 0$ at T > -3°C. This imitated the temperature dependency between -6°C and -3°C that is presented in (Phillips et al., 2018). In our parametrization, as in Qu et al. (2022), we assumed that freezing drops of
equivalent-volume diameters between 100 μm and 3500 μm contributed to SIP. The total number of splinters was calculated by multiplying Eq. (1) by the particle size distribution of freezing drops over their equivalent-volume diameters.

### 2.3 Other changes to P3

We introduced two other modifications to P3 to improve the representation of ice particle properties with nCat > 1. These modifications improve the size distributions of ice using both the one-dimensional idealized simulations (Appendix B) and in
the three-dimensional simulations (Sect. 4).

First, the collection of raindrops by small ice particles was modified to limit problems associated with the dilution of secondary ice particles. As mentioned above, P3 assumes that rain collected by ice instantaneously freezes when T < 0°C, except for a fraction that is shed, following Musil (1970). The ice–rain collection routine computes the mass ($q_{rcol}$) and the number ($n_{rcol}$)
of raindrops that are collected for every ice category based on an integration of the ice particle and raindrop size distribution multiplied by the difference in fall speed (Morrison and Milbrandt, 2015). The resulting $q_{rcol}$ and $n_{rcol}$ are then subtracted from the raindrop mass and number. The value for $q_{rcol}$ is added to the ice category responsible for the collection. When collection occurs between a small number of large raindrops and a high number of small ice crystals, the entire collected rain mass is added to the mass of the ice crystal category. Because the mass for collected rain is larger than the mass for ice crystals, the
ice crystal properties are diluted into a category with properties that resemble ice pellets. This occurs even if the resulting frozen drops would have a diameter that corresponds more closely with another category of ice. To avoid this dilution effect, we added a new routine to P3 to redirect large-collected raindrops to the most appropriate ice category when the mean mass-weighted diameter of rain is *n* times as large as the mean mass-weighted diameter of ice. Although the simulations were sensitive to the *n* variable, the best results were obtained for *n = 2*.


Second, the criteria used to add newly formed ice to an empty ice category and to merge already existing ice categories was changed to favor the presence of ice categories with small mean diameters. In P3, newly formed ice particles are added to an empty ice category if the differences between the mean mass-weighted diameters ($D_{i,m}$) of the new ice and that of the already existing populated categories are larger than a certain threshold, $\Delta D_{i,m}$. At the same time, two ice categories are merged if the
difference between their $D_{i,m}$ is < $\Delta D_{i,m}$. The choice of $\Delta D_{i,m}$ depends on the number of ice categories and the variable $\Delta D_{i,m} = 500$ μm when two ice categories are used (Milbrandt and Morrison, 2016). When simulating ice pellets, this approach results in the dilution of small ice particles when the precipitation rate is low. For example, if an existing ice category has a $D_{i,m} = 500$ μm, newly formed secondary ice particles with diameter $D_{i,m} = 10$ μm will be added to this category because the



difference between 500 µm and 10 µm is less than $\Delta_{Di,m}$. This results in the dilution of these ice splinters even if they are 50
times smaller than the mean mass diameter of the existing ice category. To avoid this problem, we modified the criteria for
merging ice categories and for adding mass to an empty ice category by using ratios instead of differences between the ice
categories $D_{i,m}$. The experiments that included this modification in our study used a threshold ratio of 10 %. Hence, two
categories merged if their $D_{i,m}$ had a relative difference of < 10 %, and new ice particles were initiated in an empty ice category
if their $D_{i,m}$ had a relative difference > 10% compared to the $D_{i,m}$ of all non-empty categories.

**2.4 Description of the conducted experiments**

Four experiments were conducted (Table 1). The control experiment (nCat1_noSIP) used the baseline version of P3 (section
2.1), with only one ice category and no SIP processes. The second experiment (nCat2_HM) used the same P3 version as in
nCat1_noSIP but with two ice categories (i.e. including HM; Morrison and Milbrandt, 2016). The third experiment
(nCat2_FFD) had two ice categories and the FFD process (i.e., no HM). The fourth experiment (nCat2_FFD_MOD), had two
ice categories, the FFD process, the modifications to the collection of rain by small ice particles, and the modifications to the
merging criteria described in Sect. 2.3.

The four experiments (Table 1) were conducted with high-resolution GEM-based hindcast simulations of an ice pellet storm
that happened in January 2020 and for which several observations of the microphysics properties are available (LT22;
Lachapelle et al. 2024). The GEM model configuration is described in the next section (3). The P3 versions used in the four
experiments were first tested with a one-dimensional model to study the effect our modifications (Sect. 2.3) in idealized
conditions (Appendix B).

Table 1: Conducted experiments

| Experiment names | Number of ice categories | SIP | Modifications to P3 (Sect. 2.3) |
|---|---|---|---|
| 1. nCat1_noSIP | 1 | No | No |
| 2. nCat2_HM | 2 | Hallett–Mossop (Milbrandt and Morrison 2016) | No |
| 3. nCat2_FFD | 2 | Fragmentation of freezing drops (L15) | No |
| 4. nCat2_FFD_MOD | 2 | Fragmentation of freezing drops (L15) | Yes |


**3. GEM model configuration and analysis**

The GEM model was used to simulate the winter storm that occurred in southern Quebec (eastern Canada) on 11 and 12
January 2020. The GEM dynamical core implicitly solves the fully compressible governing equations in time and uses a semi-





Lagrangian advection scheme (Côté et al., 1998; Girard et al., 2014). The domain was centered on Montreal (Fig. 1), within

an area of 12.0° longitude by 9.2° latitude, and with a horizontal resolution of 0.009° (≈ 1 km). The number of vertical levels was 66 with the lowest model level at ∼ 15 m above the ground. Physical parameterizations included the Interaction Soil Biosphere Atmosphere scheme (ISBA) as the surface scheme (Bélair et al., 2003, 2005), FLake as the lake scheme (Mironov et al., 2010), and the shallow convection scheme from Bélair et al. (2005). Deep convection was assumed to be resolved at that grid spacing. The simulations were driven every 1 h at the borders using the ERA5 reanalysis data (Hersbach et al., 2020),

which has a grid spacing of 31 km. The simulations were initialized with ERA5 data at 0000 UTC 10 January 2020 and were run up to 0000 UTC 14 January 2020. The entire passage of a low-pressure system in this domain was captured during this period by the simulations.

The simulated precipitation types were compared to those reported hourly at different airports in the domain and recorded in

the *Integrated Surface Database* (ISD; Smith et al., 2011). As complementary information, the probability of detection, success ratio, bias, and critical success index of rain, snow, ice pellets, and freezing rain, following Roebber (2009)., are presented in Appendix C. In general, statistics were similar among the four experiments, but the critical success index for ice pellets was better with the experiments that included the FFD SIP process.

The experiments' analysis was as follows. Hourly simulated precipitation types and precipitation rates were compared with those measured and observed at UQAM-PK, where a rain gauge measured the precipitation rate and manual observers reported the precipitation types. The precipitation types and rates that were simulated at Mirabel and Ottawa International Airports were also compared with the manual observations included in the ISD. Finally, the properties of simulated precipitation were studied. This was first done by studying cross-sections along an air parcel air parcel trajectory that ended 500 m above UQAM-

PK at 0800 UTC 12 January (Fig. 1). Second, the size distribution of precipitation were studied in an area where ice pellets were simulated at 0800 12 January 2020 (hatched area on Fig. 1). We chose to study the properties of simulated ice categories at 0800 UTC because this was in the middle of a period of continuous ice pellet precipitation measured at UQAM-PK (LT22).

## 4 Results

### 4.1 Precipitation types and phases at the surface

The four experiments produced almost the same amount of rain and snow (Figs. 2a-b, e-f, i-j, and m-n). The control experiment (nCat1_noSIP) failed to produce substantial ice pellet accumulation (Fig. 2d). Instead, an accumulation amount of > 20 mm freezing rain was simulated near Montreal (Fig. 2c). The version of P3 used in nCat1_noSIP also failed to produce solid precipitation when it was coupled to the one-dimensional model (Appendix B). However, all experiments that included SIP processes produced ice pellets at the surface (Figs. 2h,l,p). Experiments nCat2_HM, nCat2_FFD, and nCat2_FFD_MOD

simulated ice pellets along a relatively narrow latitudinal band near Montreal, spreading from west to east in the domain.





Experiments with FFD produced more ice pellets than the experiment with HM. This is consistent with the one-dimensional simulations shown in Fig. B1 in the Appendix B.

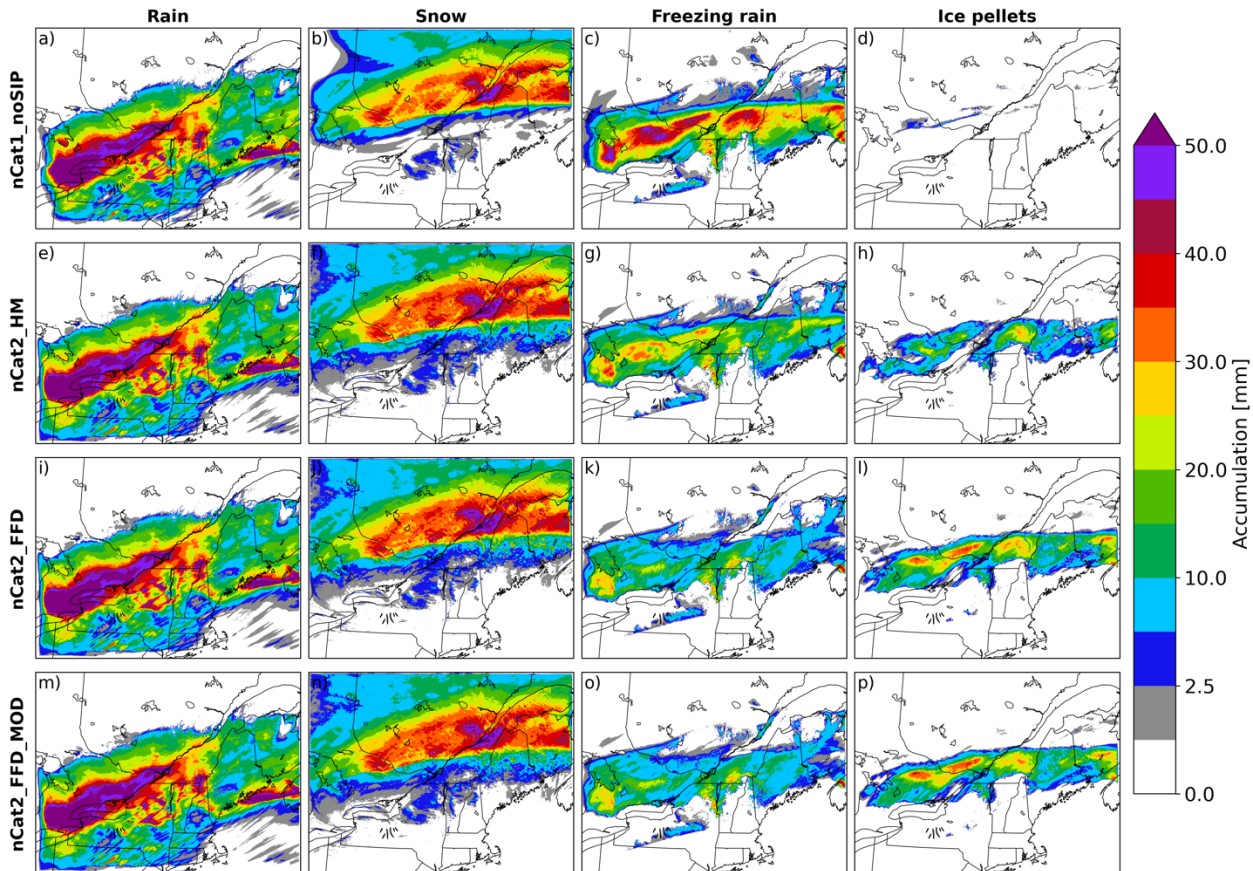

**Figure 2: Simulated accumulation between 00 UTC 10 January 2020 and 00 UTC 14 January 2020 of (a, e, i, m) rain, (b, f, j, n) snow, (c, g, k, o) freezing rain, and (d, h, l, p) ice pellets for (a–d) nCat1_noSIP, (e–h) nCat2_HM, (i–l) nCat2_FFD, and (m–p) nCat2_FFD_MOD.**


The time of the simulated precipitation type transitions was also investigated by comparing the simulated precipitation types at three sites: UQAM-PK, Mirabel International Airport, and Ottawa International Airport (Figs. 3, 4, 5, respectively). For

these sites (locations shown in Fig. 1), using two ice categories and a SIP process simulated a transition between freezing rain and ice pellets. However, the simulated transitions occurred later than the observations. These time series highlight the importance of the precipitation rate for the precipitation phase simulated at the surface. Increasing precipitation rates were associated with increasing ice pellet rates and decreasing freezing rain rates (e.g. Figs. 3c,d 09–10 UTC 12 January; Figs. 4c,d 07–08 UTC 12 January; Figs. 5c,d 08–10 UTC 12 January). These time series also demonstrated that fewer ice pellets were

simulated when the HM process was included compared to when the FFD process was included (Figs. 3b, 4b, 5b).





**Figure 3: Time series of hourly observed precipitation types (colored squares) at UQAM-PK and simulated precipitation rates from (a) nCat1_noSIP, (b) nCat2_HM, (c) nCat2_FFD, and (d) nCat2_FFD_MOD. Precipitation types are snow (blue), ice pellets (purple), freezing rain (red), and rain (green). The dashed black line shows the total simulated precipitation rate. The grey line shows the**
**precipitation rate measured by a single-alter Geonor. The observed precipitation types are from hourly manual observations (LT22). Note that between 0430 and 1600 UTC 12 January 2020, the macro photography analysis revealed the presence of tiny ice crystals (~ 200 µm) mixed with ice pellets. These were too small to be reported by manual observers.**



Figure 4: Same as Fig. 3 but at Mirabel International Airport. The manual observation were conducted hourly. The measured
precipitation rate was not available for this location.





**Figure 5: Same as Fig. 3 but at Ottawa International Airport. The measured precipitation rate was not available for this location.**

The one-dimensional model (Appendix B) confirmed the production of solid precipitation with experiments that included FFD when the minimum temperature in the subfreezing layer was < -3°C and the precipitation rate was > 0.5 mm h$^{-1}$. Therefore,

the freezing rain and ice pellet distributions at the surface were also expected to be impacted by the precipitation rate and the minimum temperature in the subfreezing layer ($T_{min}$). Figure 6 shows the fraction of solid-phase precipitation that reached the surface in the simulation domain at different hours when ice pellets were reported at UQAM-PK. Experiment nCat1_noSIP produced almost only liquid precipitation in the region with a melting layer aloft and a cold subfreezing layer of $T_{min}$ < -3°C





below (Figs. 6a–c). This region is within the red dashed lines in the panels of Fig 6. The experiments that included SIP produced
solid precipitation in a fraction of this region. As expected, the experiment using HM produced less solid precipitation than
experiments nCat2_FFD and nCat2_FFD_MOD. For those two last experiments, the phase of the precipitation was highly
correlated with the precipitation rate and $T_{min}$. At 0400 UTC 12 January 2020, the precipitation rate was low in the region
within the red dashed line and mostly liquid precipitation reached the surface (Figs. 6a, d, g, j). At 0800 and 1200 UTC, solid
precipitation reached the surface (Figs. 6h, k) in the regions characterized by higher precipitation rates (Figs. 6b, e, h, k, hatched
area). Because the SIP process cannot be activated when $T_{min} > -3°C$, solid precipitation was mostly limited to the north portion
of the red dashed line zone (Figs. 6i, l).





**Figure 6: Fraction of precipitation that reached the surface in the solid phase during three different hours, (a, d, g, j) 0400, (b, e, h, k) 0800, and (c, f, i, l) 1200 UTC 12 January 2020. The same map is presented for experiments (a–c) nCat1_noSIP, (d–f) nCat2_HM, (g–i) nCat2_FFD, and (j–l) nCat2_FFD_MOD. Each panel presents this fraction for precipitation accumulation over one hour in the simulated domain. The regions within the red dashed lines include grid points with a temperature (T) profile characterized by a melting layer with T > 0°C aloft, and a subfreezing layer with a minimum T < -3°C. Black hatch patterns indicate areas with hourly averaged precipitation rates > 2 mm h⁻¹. Orange lines shows areas with heavy hourly-averaged precipitation rates (> 7 mm h⁻¹).**





## 4.2 Hydrometeor properties along air parcel trajectories

The wind measurements recorded during the January 2020 storm indicated that a strong, low-altitude, northeasterly wind was channeled by the Saint Lawrence River Valley. LT22 suggested that the production of observed ice pellets and ice crystals in Montreal aera at that time was enhanced by the advection of ice from the northeast of Montreal where snow was reported. To investigate this hypothesis, we calculated the trajectory of an air parcel that reached an altitude of 500 m above UQAM-PK station at 0800 UTC 12 January. The altitude of 500 m corresponds to the altitude at which supercooled raindrops froze into ice pellets, as measured by the MRR-Pro installed in Montreal during the ice pellet storm (LT22). The trajectory was calculated by iteratively subtracting the distance traveled by the air parcel at 1-minute time steps. The wind values used were interpolated in time and space from the simulation outputs. The result was not sensitive to the timestep chosen and shows that trajectories ending above UQAM-PK station at an altitude of 500 m originated from the region impacted by snow to the northeast of Montreal (Fig. 1).

In this section, the properties of the simulated ice category 1 and ice category 2 are studied along a cross-section that follows the air parcel trajectory reaching UQAM-PK at 0800 UTC. As mentioned above, nCat1_noSIP was unable to produce substantial solid precipitation accumulation in the region characterized by a melting layer aloft. However, the cross section shows that a small mass mixing ratio of ice reached the surface at the north end of the melting layer (Fig. 7a). This ice resulted from the freezing of partially melted snow, making the ice particles highly rimed (Fig. 7q) and with a high mean mass-weighted diameter (Fig. 7i).





**Figure 7: Cross-sections of ice category properties simulated with the four experiments along an air parcel trajectory ending at 500 m above UQAM-PK station at 0800 UTC. The properties presented are (a–d) the ice mass of ice category 1, (e–h) the ice mass of ice category 2, (i–l) mean mass-weighted diameter of ice category 1, (m–p) mean mass-weighted diameter of ice category 2, (q–t) the rime mass fraction of ice category 1, and (u–x) the rime mass fraction of ice category 2. The latitude of the UQAM-PK station is 45.5° N and is indicated by "MTL" on the x-axis.**



At 0800 UTC 12 January 2020, nCat2_HM was similar to nCat1_noSIP, as the ice did not propagate below the melting layer (Fig. 7b). Ice category 2 was populated with a very low mass mixing ratio above the melting layer and is characterized by larger particles (Fig. 7n). In another time and space, nCat2_HM produced ice pellets at the surface (e.g. 1200 UTC, Fig. 6f).

This happened when the melting layer was colder and when the precipitation rate was higher. These conditions resulted in large ice particles (> 4000 μm) reaching the subfreezing layer and producing secondary ice through the HM process.

In the two experiments that included FFD, ice appeared in the subfreezing layer in a continuous shape that connected to the region with snow to the northeast of Montreal, at a latitude of > 46.5° N (Fig. 7). This is consistent with the hypothesis

suggesting that the advection of ice below the melting layer initiated the glaciation of the subfreezing layer and increased the area affected by ice pellets (LT22). Moreover, the latent heat released by the formation of ice pellets increased the temperature in the subfreezing layer. This is shown by the -10°C isotherm that disappeared from the subfreezing layer at latitudes < 46° N for the two experiments that included FFD (Fig. 7).

In contrast with nCat2_FFD, the two ice categories for the nCat2_FFD_MOD experiment were populated with a substantial ice mass mixing ratio in the subfreezing layer (Figs. 7d,h). The mean mass-weighted diameter of the two ice categories showed high variability in the subfreezing layer (Figs. 7l,p). Nonetheless, when one ice category contained small ice particles, the mean mass-weighted diameter of the other category was larger (Figs. 7l,p). In addition, the rime mass fraction of the smallest ice category was lower than that of largest one (Figs. 7t,x). This is consistent with our observation of small, unrimed ice

particles mixed with larger ice pellets that were 100% rimed.

### 4.3 Size distributions at the surface

The particle size distribution of the precipitation simulated at 0800 UTC 12 January 2020 were analyzed in a region where ice pellets was simulated (Figs. 1, 8). At 0800 UTC, this region had a mean precipitation rate of ~ 3.5 mm h$^{-1}$, which is the same as the mean precipitation rate between 0600 and 0800 UTC 12 January measured at UQAM-PK. The laser-optical disdrometer

measurements between 0600 and 0800 UTC 12 January were used to calculate the size distribution of ice pellets presented in Figs. 8 and B2.

At 0800 UTC 12 January 2020, nCat1_noSIP produced only liquid precipitation in the studied region. As expected, the particle size distribution of the rain species were similar to the size distribution measured by the laser-optical disdrometer (Fig. 8b).

Similarly, a large amount of rain reached the surface in this area in nCat2_HM, and the size distributions of the simulated rain are similar to that of the observed ice pellets (Fig. 8d).



At 0800, solid precipitation was simulated for the entire study domain for nCat2_FFD and nCat2_FFD_MOD. The hydrometeors simulated with nCat2_FFD were small and had a high number mixing ratio (Fig. 8f). In parallel, the ice in category two had a very low number mixing ratio. Experiment nCat2_FFD produced similar size distributions using the idealized one-dimensional model (Fig. B1c). In contrast, the size distributions with nCat2_FFD_MOD were closer to those observed in the field compared to nCat2_FFD. Similar results were also obtained with the one-dimensional model (Fig. B2d).

These results suggest that adding an efficient SIP process to the microphysics scheme can help to produce solid precipitation near the surface, but the resulting particles might have unrealistic size distribution. The modifications that were implemented in P3 in this study, which include modifications to the collection of rain by small ice particles and to the criteria for ice category merging and ice category initiation (Table 1), helped to produce more realistic size distribution of ice pellets.





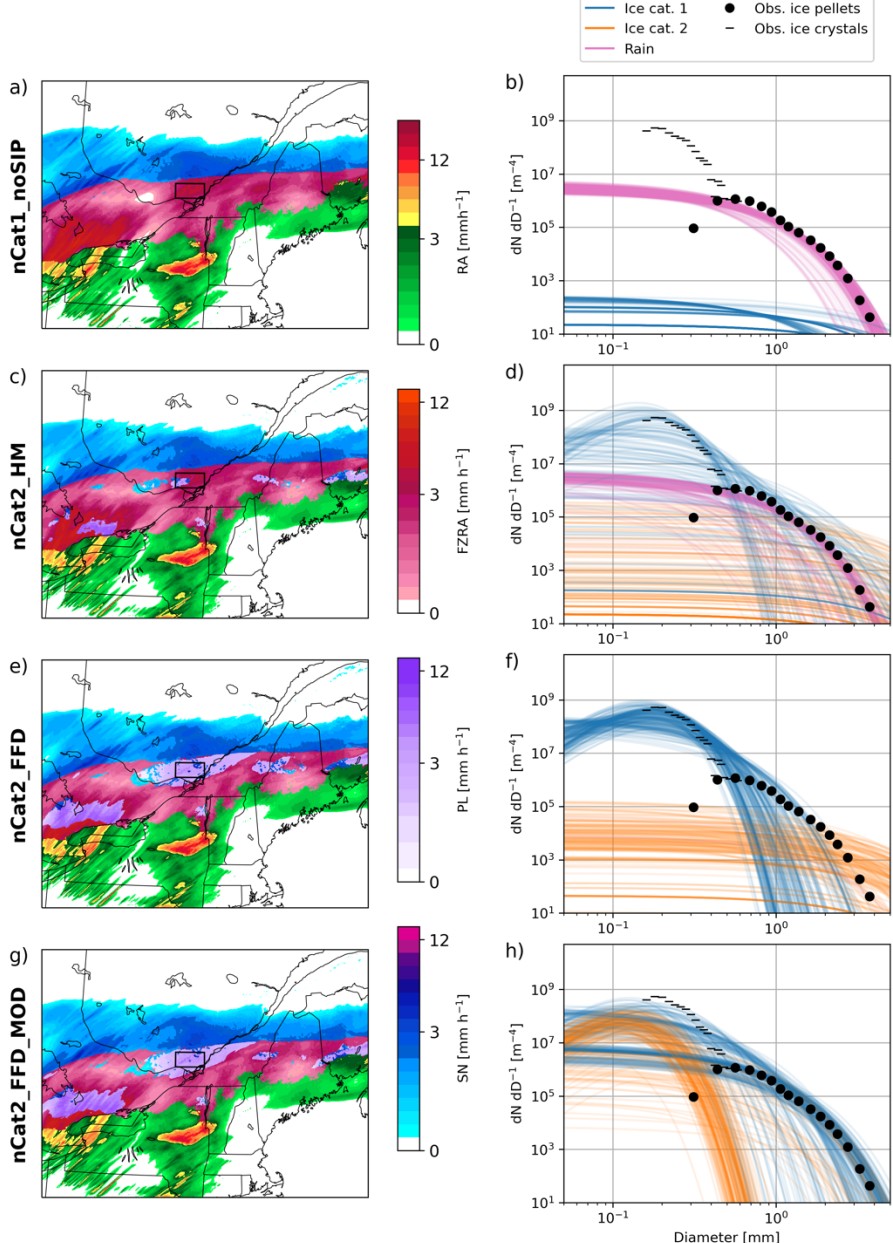

**Figure 8: (a, c, e, g) Precipitation rates of the different precipitation types at 0800 UTC 12 January 2020 by (a) nCat1_noSIP, (c)**
**nCat2_HM, (e) nCat2_FFD, and (g) nCat2_FFD_MOD. In the case of a mixture of two or more precipitation types, the type with**
**the highest precipitation rate is represented. The precipitation types are rain (RA), freezing rain (FZRA), ice pellets (PL), and snow**
**(SN) (b, d, f, h) Observed size distribution of ice pellets (black dots) and ice crystals (black horizontal lines), and size distributions**
**of rain (pink) and the two ice categories (orange and blue) simulated by (b) nCat1_noSIP, (d) nCat2_HM, (f) nCat2_FFD, and (h)**
**nCat2_FFD_MOD at the lowest model level, calculated for 200 grid points located within the study domain. The observed size**
**distribution were measured with photographs and a laser-optical disdrometer at UQAM-PK between 06 and 08 UTC 12 January**
**2020.**



**5 Conclusions**

The impacts of two secondary ice production (SIP) processes on freezing rain and ice pellet distribution at the surface were studied using high-resolution three-dimensional simulations coupled to the Predicted Particles Properties (P3) microphysics scheme.

The control experiment (nCat1_noSIP), which did not include SIP simulated mostly liquid precipitation near the surface in conditions that led to ice pellets in the field. Including the HM process (i.e., nCat2_HM) did produce some ice pellets in the experiment: a result that is not reproduced with the one-dimensional cloud model (Appendix B). The HM process produces secondary ice when large ice particles collect raindrops. Hence, this process cannot be activated if all the ice melts in the melting layer, as shown in the one-dimensional experiment. In the GEM experiment, however, some regions were characterized by partial melting aloft, fulfilling the conditions to activate the HM process. Experiment nCat2_HM simulated a smaller amount of ice pellets compared to the experiments including FFD (nCat2_FFD and nCat2_FFD_MOD).

The FFD process clearly improved the representation of ice pellets. This SIP process produced secondary ice when raindrops froze at T < -3°C. A raindrop can freeze at this temperature by collisional freezing with another frozen particle or by immersion freezing. In contrast with the HM process, conditions for the FFD process can be met when there is complete melting aloft and if secondary ice already exists in the subfreezing layer. The experiments nCat2_FFD and nCat2_FFD_MOD therefore produced ice pellets at the surface in both the idealized one-dimensional and the GEM simulations.

Adding an efficient SIP process can lead to the simulation of unrealistic ice particles size distribution. Experiment nCat2_FFD simulated a higher number of small ice particles compared to those observed in the field. By modifying the collection of raindrops by small ice particles, and the criteria for ice category merging and the ice category initiation, more realistic size distributions of ice pellets were simulated.

The properties of the simulated ice categories that were studied along the trajectory of an air parcel support the mesoscale process previously described by LT22. This process suggests that the glaciation of the subfreezing layer can be favored by the wind direction in the subfreezing layer. If the wind is strong enough, very small ice particles can be advected from the snow region to below the melting layer and cause the freezing of supercooled rain drops through collisional freezing. The secondary ice particles produced by these freezing drops have a low fall speed and can be advected further below the melting layer, increasing the area impacted by ice pellets.

Overall, our results suggest that the distributions of ice pellets and freezing rain are sensitive to these physical processes (SIPs, ice-rain collection and advection of small ice crystals) as well as to the choice of how ice categories are combined in P3. In





the future, other SIP processes, including their combination, should be tested in this context and in other weather systems. This would eventually lead to a better representation of ice pellets properties and freezing rain in models, leading to improved forecasts and climate projections. Moreover, as noted by others before us, this research shows the importance of SIP and justify the need for further laboratory experiments to improve the parametrization of SIP within atmospheric models.

**Appendix A: Immersion freezing parametrization**

The heterogeneous freezing of liquid drops and droplets in different microphysics schemes, including P3 and Thompson microphysics schemes, follows the parametrization found in Bigg (1953) (Thompson et al., 2004). For P3, this parametrization follows the equation presented in Pruppacher and Klett (2010) using the parameters for distilled water, which were experimentally identified by Barklie and Gokhale (1959). This parametrization assumes a constant concentration of ice nuclei in any given volume of liquid water. Figure A1a shows that this parametrization leads to only a small fraction of total raindrops

freezing when temperatures are > -10°C and the supercooled drops fall 1000 m at this temperature. Figure A1 also shows the same fractions when the parametrization for rainwater is used. The rainwater parameterization, which is not used in P3, would drastically increase the fraction of freezing raindrops but still not produce 100 % ice pellets, as observed on 12 January 2020.

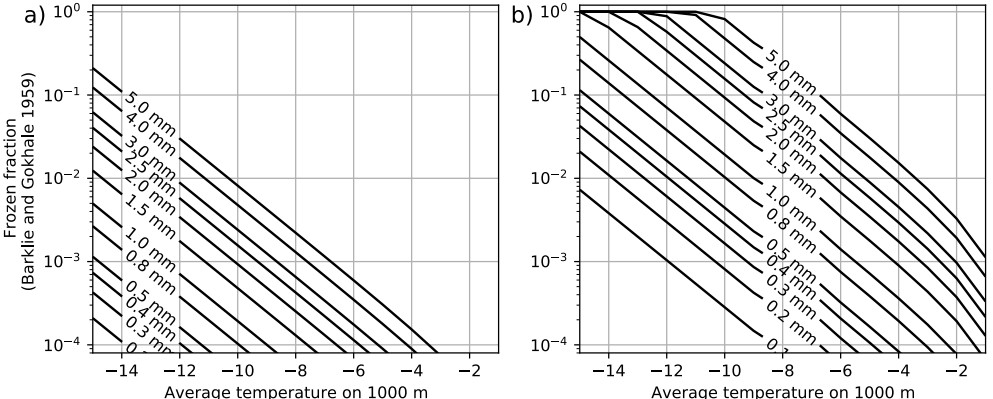

**Figure A1: The fraction of total raindrops of different diameters that freeze when falling for 1000 m, using Bigg's (1953) formulation**
**with Barklie and Gokhale's (1959) parametrization for (a) distilled water (used in P3) and (b) rainwater.**

**Appendix B: Idealized one-dimensional simulations**

**B.1 Model configuration**

One-dimensional simulations were used to study HM and FFD processes under constant atmospheric conditions. We applied a 30 s time step and 24 vertical levels (because the model top is near 4 km). The time step and the spacing between the vertical

levels were the same as for the ice pellet simulation presented in Sect. 3. Snow was initialized from the model top (4.1 km) at



a constant rate of 3.5 mm h$^{-1}$, which was the precipitation rate measured at the UQAM-PK station at 0800 UTC 12 January 2020. The vertical profiles for temperature and humidity above the UQAM-PK station were extracted from ERA5 and used as the initial conditions. The temperature profile was maintained constant during the simulations, but the humidity profile was allowed to evolve freely. To initiate freezing in the subfreezing layer, a small concentration (100 m$^{-3}$) of ice particles with a
mean mass-weighted diameter of 100 μm were introduced to the subfreezing layer between 0 m and 1000 m at the first step. Simulations were run for periods of 3 hours, after which all simulations reached a constant concentration of solid and liquid particles at every level of the column. The simulated vertical profiles of ice and rain mass and number mixing ratios, as well as particle size distributions, were compared for the different P3 versions listed in Table 1.

**B.2 Results**

nCat1_noSIP produced almost no ice in the subfreezing layer (Fig. 2). Experiment nCat2_HM also resulted in mostly liquid particles at the surface. In these two experiments, the ice mass and number mixing ratios produced at elevations below 1 km mainly came from the immersion freezing of rain (Appendix A). As the number of frozen raindrops was too small to freeze the other particles by collisional freezing, the precipitation type at the surface remained liquid. With nCat2_HM, the HM process was never triggered because the ice particles produced in the subfreezing level were not large enough.


Under the same conditions, nCat2_FFD and nCat2_FFD_MOD simulated solid precipitation at the surface. With nCat2_FFD (Figs. B1d, j), the number mixing ratio of ice quickly became high enough to freeze all raindrops below 800 m. The number of ice particles reached the maximum number allowed in P3, which is $2 \times 10^6$ m$^{-3}$ (Fig. B1j). This rapid increase is explained by the exponential behavior of SIP processes. In this experiment, the collected raindrops and the secondary ice particles were
added to the same ice category. This resulted in the simulation of a large population of small, completely rimed ice particles (Fig. B1c). Hence, the properties of the simulated precipitation particles are completely different from what was observed. In a perfect simulation, the frozen raindrops would be a similar size to the original raindrops, while the non-collected secondary ice particles would grow to form ice crystals due to vapor deposition.

With nCat2_FFD_MOD, the two modifications presented in Sect. 2.3 limited the dilution of the ice category properties and the two ice categories were more realistically populated in the subfreezing layer (Figs. B1e, k). At the level closest to the surface, the size distributions of the two ice categories were consistent with those observed (Fig. B2d; LT22). The ice properties for category two were likely those of ice pellets, with a size distribution similar to that of raindrops simulated at 1000 m (Fig. B2d) and the density of bulk ice (900 kg m$^{-3}$) (Fig. B1e). The ice properties for category one were similar to the observed ice
crystals. The particles were not rimed (Fig. B1e) and were small (Fig. B2d). However, the simulated ice crystals were smaller than those observed. That might be explained by the limited water vapor that was available in the idealized one-dimensional simulations, where the relative humidity rapidly reached saturation over ice in the subfreezing layer.



In addition to performing simulations with the observed precipitation rate and temperature profile, sensitivity tests were
conducted by varying the precipitation rate and the minimum temperature in the subfreezing layer with the P3 version in
nCat2_FFD_MOD. These tests showed that the phase of precipitation after the 3-hour simulations was sensitive to these
parameters, resulting in solid precipitation at the surface when the minimum subfreezing temperature was < -3°C and when
the precipitation rate was > 0.5 mm h$^{-1}$. In contrast, freezing rain was simulated at the surface when the temperature in the
subfreezing layer was warmer or the precipitation rate was lower. These conditions produced secondary ice particles that were
insufficient to initiate the process leading to the freezing of all supercooled raindrops.

**Figure B1: Vertical profiles of (a, g) temperature (solid black line) and final dew-point temperature (dashed red line) of
nCat2_FFD_MOD, (b–e) total ice mass mixing ratio (in g kg$^{-1}$) for category 1 (blue dots) and category 2 (orange dots), rimed ice
mass mixing ratio for category 1 (blue plus signs) and category 2 (orange plus signs), and (h–k) total ice number mixing ratios of ice
from category 1 (blue dots), category 2 (orange dots), and rain (pink dots). Results generated after 3 h of constant precipitation rate
at the model top of 3.5 mm h$^{-1}$ for (b, h) nCat1_noSIP, (c, i) nCat2_HM, (d, j) nCat2_FFD, and (e, k) nCat2_FFD_MOD. Nmax is 2
× 10$^6$ m$^{-3}$.**





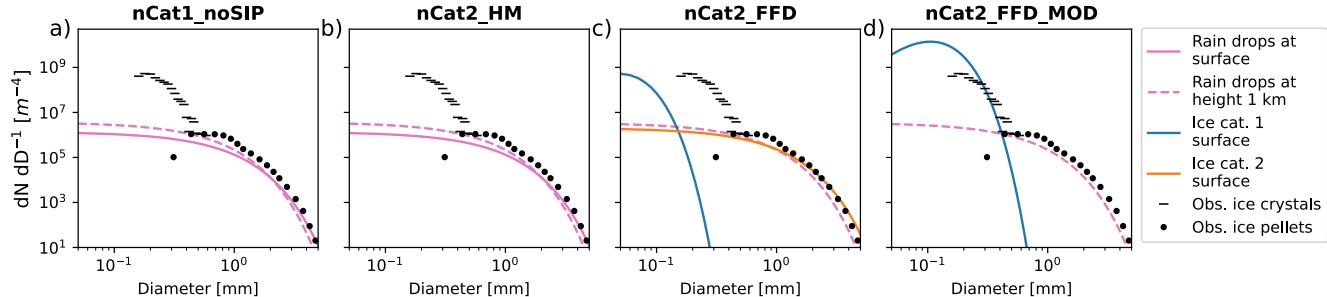


**Figure B2: Size distributions of simulated rain (pink) and ice (blue for category 1 and orange for category 2) at the lowest vertical level (solid line) and at a height of 1 km (dashed line) after 180 minutes for (a) nCat1_noSIP, (b) nCat2_HM, (c) nCat2_FFD, and (d) nCat2_FFD_MOD. Size distributions are compared with ice pellets (black dots) and ice crystals (dashed black) measured at UQAM-PK between 0600 UTC and 0800 UTC January 2020 (LT22).**

**Appendix C: Precipitation type statistics**

Indices for a performance diagram were calculated based on the simulated and observed hourly precipitation types at the different airports in the simulation domain. The performance diagram presented in Fig. C1 follows the method presented by Roebber (2009). The diagram includes the four precipitation types considered in our study (rain, snow, ice pellets and freezing rain). For the four simulations, the diagram shows the Probability of Detection (POD), the Success Ratio (SR), the false alarm

ratio (FAR), and the critical success index (CSI). Overall, this analysis shows that the critical success ratio for ice pellets improves drastically when SIP processes are added to P3. The critical success ratio for the three other precipitation types do not vary much but do show improvement. A threshold of 0.2 mm h$^{-1}$ was used to filter out low simulated precipitation rates that may not be observed by manual observers. Choosing a lower precipitation rate threshold results in an increase of the probability of detection and a decrease of the success ratio, but does not affect the general conclusion of this Appendix.






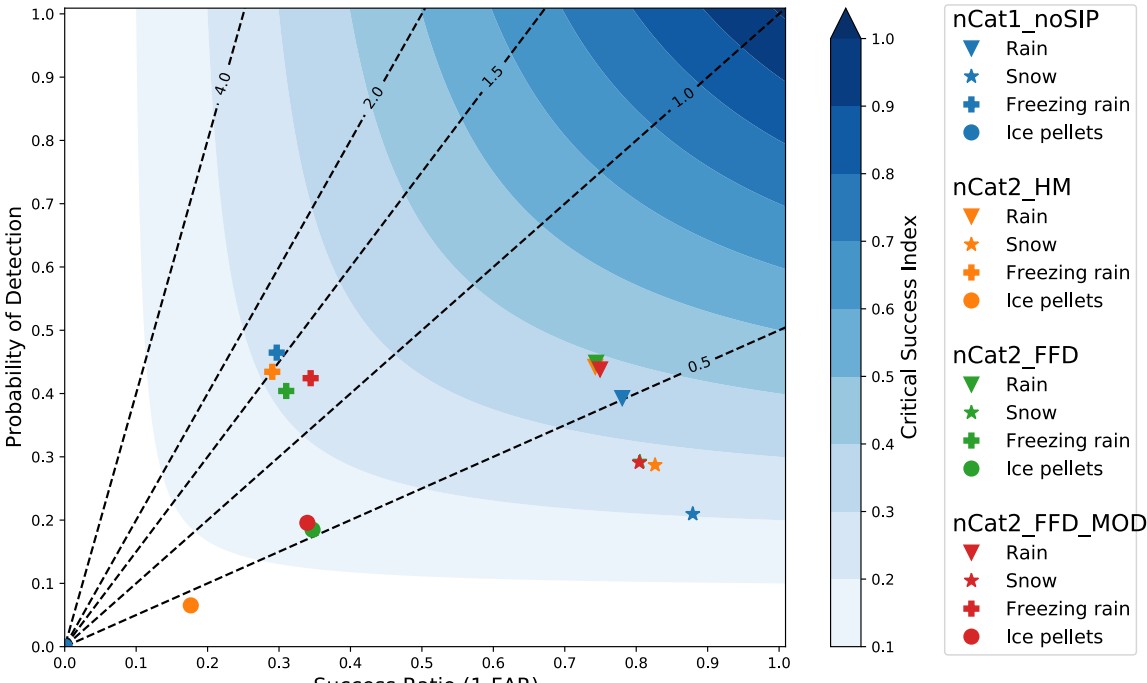

**Figure C1: Performance diagram as presented by Roebber (2009). The Probability of detection (POD), the success ratio (SR) and the critical success index were calculated for four precipitation types (shapes) and four simulations (colors) using the hourly observations of precipitation types from the Integrated Surface Database (Smith et al., 2011) and hourly simulated precipitation types at the surface. Only precipitation types with an hourly water equivalent accumulation > 0.2 mm were considered. Dashed lines show the slope POD/SR.**


**Data availability**

Observational data for the 12 January storm are available online on The Canadian Dataverse Repository Borealis (https://borealisdata.ca/dataset.xhtml?persistentId=doi:10.5683/SP3/TGS5HU). Simulation outputs are available upon

request.

**Code availability**

The code producing the figures is available upon request.

**Competing interests**

The authors declare no competing interests.



**Acknowledgements**

The authors would like to thank Hadleigh Thompson, Charlie Hébert-Pinard, Karel Veilleux, Aurélie Desroches-Lapointe, Félix Biron, Sébastien Marinier, Julien Chartrand, and Sara-Ann Piscopo for assistance with manual observations during the 11–12 January 2020 ice pellet storm. We also wish to thank François Roberge and Katja Winger for their assistance with the simulations that were conducted on infrastructure from Digital Research Alliance of Canada, whom we also thank for their

support. We would like to thank Tangui Picart for his constructive comments. Financial support for this study was provided by the Fonds de recherche du Québec Nature et Technologies, Natural Sciences and Engineering Research Council of Canada, Canada Foundation for Innovation, and the Canada Research Chairs program (CRC-2018-00312).

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
