# Peer review of "Effect of Secondary Ice Production Processes on the Simulation of ice pellets using the Predicted Particle Properties microphysics scheme"

_EGUsphere, 2024_

## Editor Comment (EC1)

Review of egusphere-2024-594: 'Effect of Secondary Ice Production Processes on the Simulation of ice pellets using the Predicted Particle Properties microphysics scheme'

The manuscript describes forecast experiments for a winter storm event that produced snow, freezing rain, and ice pellets. The forecast model does a remarkable job of reproducing the precipitation rates at the comparison site, which could in part be a reflection of the system being in a more predictable regime of synoptic forcing. The experiments conclude that the P3 scheme is able to produce realistic ice pellet accumulations with a parameterization of small ice production by shattering of freezing drops. I think this is nice study that presents a problem and a physically plausible solution. I have a just a few questions to clarify some points.

Minor Comments:

1. Line 112: 'Immersion freezing of cloud droplets and rain can occur when T < −4C, following the volume and temperature dependent formulation presented in Bigg (1953)'

From Bigg's plot, highest practical mean temperature for freezing is about −20C for very large drops. I would not expect any freezing from that paramterization at −4C, but I'm not familiar with the Barklie and Gokhale (1959) formulation. Does P3 really produce much drop freezing from this process at temperatures higher than, say, −15C? Freezing by ice crystal capture could happen, of course. Also, Bigg 1953 used distilled water, so I consider it to be something of a lower limit for homogeneous freezing.

2. Line 139: 'collect 1 mg of rain'

Should 'rain' be 'droplets'? Studies of the HM process have used small droplet riming, which probably does not apply to collection of larger rain-sized drops.

3. Concerning the modified merging criteria: Clearly this change produces a more realistic result in this case, but I'm curious if this was also tested on other types of convection to see if has any adverse affects.

4. Line 217: 'but the critical success index for ice pellets was better with the experiments that included the FFD SIP process'

I think calling it 'better' is an understatement. The base simulation has essentially zero CSI, and the differences with HM and then FFD are quite substantial and worth saying a bit more about.

5. Line 415: 'Snow was initialized from the model top' What were the values of number concentration, mass mixing ratio, etc. for the snow? What was the mean diameter of the particles?

6. Figure B1: The rain number by height seems to drop about an order

of magnitude, which would be more than accounted for by air density change over 2km. Are there processes that are affecting the rain number (such as the liquid fraction)?

7. Figure B1: Why do FFD and FFD-MOD have abrupt melting of all ice compared to the first two (e.g., blue circles at 2-3km in panel i vs. panel j)?

8. Line 454: 'or the precipitation rate was lower'

Can you elaborate very briefly on how the precipitation rate affected freezing? Is it simply an insufficient production of ice crystals such that collection was too small to initiate freezing?

9. Fig. B1: Would it be possible to plot the number concentration as number/m3 so that it is easier to compare different altitudes? Not a big deal if not. (Likewise in the upper row, mass content would be easier to compare vertically than mixing ratio.)

10. Are panels B1a and B1g the same plot? If so, please make that clear. Or perhaps there is something else that could replace the second one.

11. Fig. B1 (one more thing!): Does 'rain' here represent only qr (and nr) or does it include liquid mass on ice? Can you represent the liquid fraction on ice here, perhaps as a line plot in the range of 0-1?

Editorial comments:

1. Line 239: 'This is consistent with our observation'

Is this referring to the "UQAM-PK weather station in downtown Montreal"? If so, that could be made clear. Or is it a personal observation (if so, at what approximate times)?

2. Line 344: 'The hydrometeors simulated with nCat2-FFD were small and had a high number mixing ratio (Fig. 8f)'

This is specifically for ice species 1, correct? Not all hydrometeors. The following sentence would then make more sense. (Although instead of "In parallel" I would suggest 'Conversely' or something that indicates opposite characteristics for ice2.)

3. Line 434: 'In this experiment'

I suggest clarifying this as 'In experiment nCat2FFD' to avoid confusion since two experiments are stated in the first sentence.

4. Line 437: 'a similar size to the original raindrops' And what size are they?

---

## Author Comment (AC1)

**Answers to the Anonymous Referee 1**

*Answers are in italic blue text.*

**RC1: 'Comment on egusphere-2024-594', Anonymous Referee #1, 28 Mar 2024**

Review of "Effect of Secondary Ice Production Processes on the Simulation of ice pellets using the Predicted Particle Properties microphysics scheme" by Lachapelle et al.

The authors of this manuscript examined the effect of secondary ice production (SIP) processes on the simulation of ice pellets using an NWP model with a double moment bulk microphysical scheme (P3). Both Hallett-Mossop (HM) and fragmentation of freezing drops (FFD) processes are examined. It is found that adding HM or FFD would significantly improve the simulation of ice pellets for this specific case (both full simulation and 1D idealized simulation). I enjoyed reading this manuscript which is well written, and the results are clearly presented. I made several suggestions below that may help.

*Thank you for your comments.*

**Main comments:**

1.      L134-136: I believe that the modification of the default HM parameterization in P3 should not be a major obstacle to examining the combined effect of HM and FFD. The interaction between HM and FFD could be quite complex and sensitive to how both HM and FFD are parameterized, which could be a continued study on it alone. Nevertheless, I think it would be worth mentioning in the manuscript about this complexity which warrants further studies.

*Preliminary tests (not shown here) have demonstrated that adding HM to a configuration of P3 that includes FFD does not change significantly the accumulation of ice pellets. In our parametrization of these processes, both HM and FFD occur under similar conditions, when supercooled drops are collected by ice particles. Hence, because the parametrization of FFD is more efficient than the parametrization of HM, adding HM to FFD did not have much impact.*

*The sentences at lines 134-136 included too many details that were possibly confusing and misleading for the reader. The original sentences at lines 134-136 was:*

*"Observations collected in the field suggest that FFD and HM were active SIP processes during the 12 January 2020 ice pellet episode (LT22; Lachapelle et al., 2024).The two SIP processes were analyzed individually; no simulation used more than one SIP process simultaneously because this would result in the production of secondary ice through two distinct processes freezing the same raindrop and would require the default implementation of HM to be modified."*

*For clarity, the sentences will be changed to (changes are in bold):*

*"Observations collected in the field suggest that FFD and HM were active SIP processes during the 12 January 2020 ice pellet episode (LT22; Lachapelle et al., 2024). **However, because the objective of this work is to examine how SIP affects simulated precipitation types and particle size distributions, the two SIP processes, HM and FFD, were used individually. This approach facilitated the understanding of their respective effects.**"*

*We will also add the following sentence mentioning future studies in the conclusion (near line 395):*

*"Future research should also include simulations combining multiple SIP processes, from which could emerge complex interactions and feedback processes."*

2.      L137-138/L428-429: the default HM parameterization in P3 scheme with a threshold of 4000 µm for the mean-mass D of ice particles seems extremely large, e.g. some graupels could be much smaller than 4000 µm. Some of the previous studies have disregarded this threshold or using a smaller one. Would it be possible to test different thresholds which might have significant impacts on the results?

*Different studies used different ice diameter thresholds to parametrize HM (e.g., Sullivan et al., 2018; Sotiropoulou et al., 2021, 2020; Qu et al., 2022). An ice diameter of 4000 µm is large and limits SIP compared to using lower thresholds. Since the submission of the manuscript, the default threshold diameter for HM in P3 has been changed to 1000 µm (Cholette et al., 2024; based on Qu et al., 2022). For this reason, the threshold will be changed to 1000 µm in the revised version of the manuscript in all simulations (1D and 3D).*

*As expected, lowering the threshold increased the amount of ice pellets and decreased the amount of freezing rain (Figs. R1, R2, R3). Although Fig. R3 suggests an improvement in ice pellet statistics for this case, the region of accumulated ice pellets and the simulated particle size distributions are similar to nCat2_HM in the submitted manuscript. Hence, the main conclusions of this study are the same.*

*The following sentences will be added in the methodology section (line 140):*

*"Different ice diameter thresholds were used in different studies to activate HM (e.g., Cholette et al., 2024; Qu et al., 2022; Sotiropoulou et al., 2021; Sullivan et al., 2018). Sensitivity tests showed that the accumulated amounts of ice pellets and freezing were sensitive to this value (e.g., ice pellet amounts decreased with a larger ice diameter threshold)."*

*And in the conclusion (near line 395):*

*"The simulations are sensitive to the parametrization of SIP. For example, increasing the ice diameter threshold for HM decreases the amount of ice pellets produced. The identification of an optimal SIP parameterization for ice pellet and freezing rain simulation will require more observations and modeled cases."*

[Figure]

*Fig. R1. Ice pellet accumulation simulated with (a) nCat2_HM configuration that used an ice diameter threshold of 4000μm, (b) nCat2_HM1mm configuration that used an ice diameter threshold of 1000μm, and (g,h,i) their differences.*

[Figure]

*Fig. R2. Same as Fig. R1 but for freezing rain.*

[Figure]

*R3. Same as Fig. C1 in the submitted manuscript with the new simulation nCat2_HM1mm. In the revised manuscript, the simulation nCat2_HM will be replaced by nCat2_HM1mm.*

3.      L432-433: the maximum number allowed ($2\times10^6$ m$^{-3}$) for ice number concentration seems quite small. In situ data suggests that much larger values are possible even without counting those ice particles smaller than ~50 µm. As SIP will produce a large amount of tiny ice splinters, the number concentration might peak locally at a high value. Although the exact maximum value is arguable, $2\times10^6$ m$^{-3}$ seems definitively too low. This means some large Ni will be automatically clipped at this lower value and the total Ni is therefore reduced. I'm wondering if the author tested other thresholds and whether the results are significantly different.

*Gultepe et al. (2015)[1] mentioned that ice can reach a number concentration $> 10^6$ m$^{-3}$. Girard and Blanchet (2001)[2] suggest that ice fog number concentration is always $< 4x10^6$m$^{-3}$. Hence, we think that an upper limit of $2\times10^6$ m$^{-3}$ is realistic in most cases, but we agree that it might be too small under some circumstances.*

*The concentration of ice reached the limit of $2\times10^6$ m$^{-3}$ in the experiments nCat2_HM and nCat2_FFD but not in the experiments nCat1_noSIP and nCat2_FFD_MOD. nCat2_FFD_MOD did not simulate such a high concentration because the modifications added limited the ice multiplication to realistic concentrations; the observed concentration of ice crystals was estimated to be between $1x10^4$ and $1x10^5$ m$^{-3}$ during the ice pellet storm presented in this study[3]. Increasing the limit could result in more ice pellets produced by experiments nCat2_HM and nCat2_FFD. Sensitivity studies could be conducted in the future to explore the impacts of modifying this limit for other winter events in which SIP processes play an important role.*

4.      L169: could the authors describe more about the simulation results of using more than 2 ice categories? My understanding is that with more ice categories, the different sizes of ice particles should be better represented. Although for many reasons, such as our limited knowledge of SIP, etc. a better physical model might not produce better prediction results. I believe more discussion on this would be helpful.

*At line 169, the variable "n" does not refer to the number of ice categories but rather to a new parameter that we introduced. To avoid this confusion, we will improve the following lines (166-169):*

*"To avoid this dilution effect, we added a new routine to P3 to redirect large-collected raindrops to the most appropriate ice category when the mean mass-weighted diameter of rain is n times as large as the mean mass-weighted diameter of ice. Although the simulations were sensitive to the n variable, the best results were obtained for n = 2."*

[1] I. Gultepe et al., « A review on ice fog measurements and modeling », *Atmospheric Research* 151 (2015): 2-19, https://doi.org/10.1016/j.atmosres.2014.04.014.

[2] Eric Girard et Jean-Pierre Blanchet, « Microphysical Parameterization of Arctic Diamond Dust, Ice Fog, and Thin Stratus for Climate Models », *Journal of the Atmospheric Sciences* 58, nᵒ 10 (1 mai 2001): 1181-98, https://doi.org/10.1175/1520-0469(2001)058<1181:MPOADD>2.0.CO;2.

[3] M. Lachapelle et J. M. Thériault, « Characteristics of precipitation particles and microphysical processes during the 11–12 January 2020 ice pellet storm in the Montréal area, Québec, Canada », *Monthly Weather Review*, 2022, 1043-59, https://doi.org/10.1175/mwr-d-21-0185.1.

*to this revised sentence:*

*"To avoid this dilution effect, we added a new routine to P3 to distribute large-collected raindrops to the most appropriate ice category when the mean mass-weighted diameter of rain is twice as large as the mean mass-weighted diameter of ice."*

*Concerning using P3 with more than two ice categories, the results of one-dimensional simulations with 3 and 4 ice categories are shown in Figs. R4-R7. The results, including the PSDs, are similar to those obtained with two ice categories. The authors' hypothesis is that two ice categories are enough to simulate winter precipitation types and processes, as particles remain relatively small. During convective weather, however, a higher number of ice categories may be necessary as hail sizes' range can be wide. This hypothesis will be pursued in further studies.*

*A sentence will be added to the methodology, after the description of the conducted experiments (line 197):*

*"One-dimensional simulations were also performed using three and four ice categories. Similar results were obtained with these simulations (not shown) compared to those obtained using two ice categories, suggesting that two ice categories are enough to represent the precipitation types and properties observed during this ice pellet storm."*

[Figure]

*Fig. R4. Same as Fig. B1 but with 3 ice categories.*

[Figure]

*Fig. R5. Same as Fig. B2 but with 3 ice categories.*

[Figure]

Fig. R6. Same as Fig. B1 but with 4 ice categories.

[Figure]

Fig. R7. Same as Fig. B2 but with 4 ice categories.

5.      Figure 3-5: it seems the best results from nCat2_FFD_MOD still overestimated the period of freezing rain compared to the observation, particularly for UQAM-PK. Might this suggest that current SIP rate in this study is not fast enough to convert liquid into ice?

*Yes, it might. As shown with the experiment in which we reduced the ice diameter threshold for HM (i.e., answer to your comment #2), the accumulated freezing rain and ice pellets are sensitive to how SIP is parameterized. We only based our comparisons on the observed types of precipitation (temporal evolution mainly) and PSDs at a specific location. Although we show*

*that the modified nCat2_FFD_MOD better reproduced ice pellets, we think that more cases and more observations are needed to improve the parameterizations.*

*We will add the following comment in section 4.1:*

*"In addition, all the experiments produced fewer hours of ice pellet than those observed. This suggests that increasing the efficiency of SIP could decrease the difference between simulated and observed precipitation types. However, more cases and observations are needed to improve the parameterizations."*

6.      Fig. B1: nice results from the 1D simulation which illustrates well the impact of modifying the FFD process (section 2.3), e.g. rime ice with similar size to raindrops + much smaller ice crystals!

*Thank you!*

7.      Fig. B2d: the ice cat 2 (orange line) is missing.

*Thank you for noticing this. Here is the corrected figure.*

[Figure]

*Fig. R8. Corrected Fig. B2.*

---

## Author Comment (AC2)

**Answers to the Anonymous Referee #2**

*Answers are in italic blue text.*

**RC2: 'Comment on egusphere-2024-594', Anonymous Referee #2, 22 Apr 2024**

1.      I agree with the review posted by Reviewer #1. My only additional comment would be that Figure 2 and its related discussion would be aided with some ground truth observations of precipitation type within the shown domain. For example, all snow observation locations would be plotted in the "snow" column of subplots. These observations would ideally come from ASOS, LSR, or crowdsourced mPING reports. Without these observations, it is difficult to determine whether the simulations are improved with the SIP inclusion across the domain (and not just at the small domain of subsequent analyses).

*Thank you for your comments.*

*We only have access to a few stations that reported hourly accumulated precipitation, most of them are not available. However, we will add the following figure (Fig. R1, which will become Fig. 3 in the revised manuscript) showing the observed and simulated number of hours for rain, snow, freezing rain, and ice pellets.*

*A sentence referring to this figure will be added to the main text (line 237):*

*"Finally, the total number of hours during which snow, rain, freezing rain, and ice pellets were simulated with the experiments that included the FFD process were similar to the observations included in the ISD (Fig. 3)." (ISD: Integrated Surface Database)*

[Figure]

*Fig. R1: New Figure 3. Total number of hours during which (a) rain, (b) snow, (c) freezing rain, and (d) ice pellets were reported at airports included in the ISD database between 00 UTC 10 January 2020 and 00 UTC 14 January 2020. (e-t) Total number of hours during which the simulated precipitation type was (e, i, m, q) rain, (f, j, n, r) snow, (g, k, o, s) freezing rain, and (h, l, p, t) ice pellets for (e–h) nCat1_noSIP, (i–l) nCat2_HM, (m–p) nCat2_FFD, and (q–t) nCat2_FFD_MOD. Only precipitation types with a rate > 0.2 mm h⁻¹ were considered.*

---

## Author Comment (AC3)

*Answers are in italic blue text.*

**RC3: 'Comment on egusphere-2024-594', Anonymous Referee #3**

Processes on the Simulation of ice pellets using the Predicted Particle Properties microphysics scheme' The manuscript describes forecast experiments for a winter storm event that produced snow, freezing rain, and ice pellets. The forecast model does a remarkable job of reproducing the precipitation rates at the comparison site, which could in part be a reflection of the system being in a more predictable regime of synoptic forcing. The experiments conclude that the P3 scheme is able to produce realistic ice pellet accumulations with a parameterization of small ice production by shattering of freezing drops. I think this is nice study that presents a problem and a physically plausible solution. I have a just a few questions to clarify some points.

*Thank you for your comments.*

**Minor Comments:**

1. Line 112: 'Immersion freezing of cloud droplets and rain can occur when T < -4C, following the volume and temperature dependent formulation presented in Bigg (1953)' From Bigg's plot, highest practical mean temperature for freezing is about -20C for very large drops. I would not expect any freezing from that parametrization at -4C, but I'm not familiar with the Barklie and Gokhale (1959) formulation. Does P3 really produce much drop freezing from this process at temperatures higher than, say, -15C? Freezing by ice crystal capture could happen, of course. Also, Bigg 1953 used distilled water, so I consider it to be something of a lower limit for homogeneous freezing.

*With the parametrization of immersion freezing used in P3, from Bigg (1953), we have calculated that <0.1% of raindrops with diameters of 1 mm would freeze at temperatures >-10°C during a fall of 1000 m. This calculation is presented in the Appendix A. As you mention in your comment, this is negligible. In the one-dimensional simulation, this process leads to the freezing of a minority of raindrops. In the absence of SIP with nCat1_noSIP simulation, this process explains the production of ice with a mass mixing ratio $<10^{-5}$ g m$^{-3}$ (Fig. R1b,f).*

*To avoid confusion, the word "small" will be changed for "negligible" in the following sentence (line 112):*

*"In Appendix A, we show that this parametrization of immersion freezing leads to the freezing of a  **negligible** fraction of raindrops in the atmospheric conditions observed during ice pellet events."*

2. Line 139: 'collect 1 mg of rain' Should 'rain' be 'droplets'? Studies of the HM process have used small droplet riming, which probably does not apply to collection of larger rain-sized drops.

*Sensitivity studies of including cloud droplets in the HM process in P3, including its combination with raindrops, are shown in Cholette et al., (2024). It was found that the overall statistics of freezing rain remain similar when only cloud droplets are included in HM compared to raindrops. Since Cholette et al. (2024)[1] used the same atmospheric model and microphysics, with similar grid spacing, we think that their conclusions will apply to this case as well.*

3. Concerning the modified merging criteria: Clearly this change produces a more realistic result in this case, but I'm curious if this was also tested on other types of convection to see if has any adverse affects.

*No, it has not been tested yet, this is part of future studies, but we agree with the reviewer that it may also change properties of hail-type hydrometeors.*

*We will add a sentence in the last paragraph of the conclusion concerning this (near line 395):*

*"Finally, potential adverse effects of the modifications presented in this work should be studied in other types of weather, including hail formation during severe summer weather."*

4. Line 217: 'but the critical success index for ice pellets was better with the experiments that included the FFD SIP process' I think calling it 'better' is an understatement. The base simulation has essentially zero CSI, and the differences with HM and then FFD are quite substantial and worth saying a bit more about.

*Good points, the following sentence (L217-218):*

*"In general, statistics were similar among the four experiments, but the critical success index for ice pellets was better with the experiments that included the FFD SIP process."*

*will be changed to:*

*"For rain and snow, the critical success index was slightly improved for the simulations including SIP. For ice pellets, adding SIP clearly improved the critical index because the baseline simulation produced a negligible amount of this precipitation type. The two simulations that included FFD reached the highest critical success ratio. For freezing rain, adding SIP decreased slightly the probability of detection. However, for the simulation including FFD and our modifications, the probability of detection decreasing was counterbalanced by a slight success ratio increase."*
* * *
[1] Cholette, M., Milbrandt, J. A., Morrison, H., Kirk, S., and Lalonde, L.-É.: Secondary Ice Production Improves Simulations of Freezing Rain, Geophysical Research Letters, 51, e2024GL108490, https://doi.org/10.1029/2024GL108490, 2024.

5. Line 415: 'Snow was initialized from the model top' What were the values of number concentration, mass mixing ratio, etc. for the snow? What was the mean diameter of the particles?

*In a first step, the variables $N_i$ and $Q_i$ that were prescribed at the one-dimensional model's highest level were the same as those simulated by the three-dimensional model at 4 km MSL at locations where the precipitation rate was 3.5 mm $h^{-1}$. However, initiating the highest level with these values resulted in a precipitation rate of 1 mm $h^{-1}$ in the one-dimensional model. Hence, we multiplied these values of $N_i$ and $Q_i$ by 3, which resulted in a precipitation rate of 3.5 mm $h^{-1}$. The mean mass-weighted diameter of the initiated snow was approximately 1 mm.*

*The following sentences will be added (line 417):*

*"The mass and number mixing ratio, $Q_i$ and $N_i$, of the snow initiated at the model's highest level were chosen to reproduce the observed precipitation rate of 3.5 mm $h^{-1}$. To do so, $Q_i$ and $N_i$ were first extracted from the three-dimensional simulations at 4 km MSL at locations where the precipitation rate was 3.5 mm $h^{-1}$. Then, the values of the extracted $Q_i$ and $N_i$ had to be modified by a factor of three to obtain the observed surface precipitation rate."*

6. Figure B1: The rain number by height seems to drop about an order of magnitude, which would be more than accounted for by air density change over 2km. Are there processes that are affecting the rain number (such as the liquid fraction)?

*The number concentration of raindrops decreases from around 1800 $m^{-3}$ at 2600 m MSL to around 500 $m^{-3}$ at the lowest level. Investigations of the different processes activated in P3 revealed to this was almost entirely due to rain self-collection.*

7. Figure B1: Why do FFD and FFD-MOD have abrupt melting of all ice compared to the first two (e.g., blue circles at 2-3km in panel i vs. panel j)?

*Thank you for noticing this result. We found an error in the initiation of the vertical layers, which are needed for sedimentation, in the one-dimensional model used in the submitted manuscript. This error caused an unrealistic slow melting for experiments nCat1_noSIP and nCat2_HM.*

*The experiments were conducted again with a corrected script by using 41 evenly spaced vertical model levels up to 4 km MSL. All levels had a depth of near 100 m. Overall, these new simulations produced similar results to those presented in the submitted manuscript (Figs. R11-R12) and the unrealistic melting process that was previously simulated with nCat1_noSIP and nCat2_HM was not reproduced. Since the temperature increases rapidly in the melting layer, the new results presented in the figures below, show that ice completely melts after 2 or 3 vertical levels and before reaching T=1.5°C.*

[Figure]

Fig. R1: Revised Fig. B1.

[Figure]

Fig. R2: Revised Fig. B2.

8. Line 454: 'or the precipitation rate was lower' Can you elaborate very briefly on how the precipitation rate affected freezing? Is it simply an insufficient production of ice crystals such that collection was too small to initiate freezing?

*More context will be given to these tests at the beginning of this paragraph (new sentences are in bold):*

*"In addition to performing simulations with the observed precipitation rate and temperature profile, sensitivity tests were conducted by varying the precipitation rate and the minimum temperature in the subfreezing layer with nCat2_FFD_MOD.* **First, given the temperature**

*threshold in our FFD parametrization, it was expected that secondary ice would not be produced in warmer conditions. Second, the FFD parametrization depends strongly on freezing raindrops diameter. Hence, higher precipitation rates are expected to produce larger raindrops, producing more ice particles. As expected, … [rest of the paragraph]"*

9. Fig. B1: Would it be possible to plot the number concentration as number/m3 so that it is easier to compare different altitudes? Not a big deal if not. (Likewise in the upper row, mass content would be easier to compare vertically than mixing ratio.)

*It has been changed. See the revised Figures R1 and R2 above.*

10. Are panels B1a and B1g the same plot? If so, please make that clear. Or perhaps there is something else that could replace the second one.

*They are the same plots. Panel g is deleted to avoid duplicates (see revised Fig. R1).*

11. Fig. B1 (one more thing!): Does 'rain' here represent only qr (and nr) or does it include liquid mass on ice? Can you represent the liquid fraction on ice here, perhaps as a line plot in the range of 0-1?

*Qr and Nr are rain mass and number. The mass of the liquid on ice is included in the total ice mass mixing ratio, Qi. We added the liquid mass on ice (Qi,liq) in Figure B1 (now revised Fig. R1; small vertical bars).*

Editorial comments:
1. Line 239: 'This is consistent with our observation' Is this referring to the "UQAM-PK weather station in downtown Montreal"? If so, that could be made clear. Or is it a personal observation (if so, at what approximate times)?

*This will be clarified by modifying the following sentence (changes are in bold):*

*"This is consistent with our observation **conducted at UQAM-PK of small**, unrimed ice particles mixed with larger ice pellets that were 100% rimed (**LT22**)."*

2. Line 344: 'The hydrometeors simulated with nCat2-FFD were small and had a high number mixing ratio (Fig. 8f)' This is specifically for ice species 1, correct? Not all hydrometeors. The following sentence would then make more sense. (Although instead of "In parallel" I would suggest 'Conversely' or something that indicates opposite characteristics for ice2.)

*To avoid this confusion, we will modify these two sentences for (changes are in bold):*

*"The hydrometeors simulated with nCat2_FFD **for ice category one** were small and had a high number mixing ratio (Fig. 8f). In **contrast**, the ice in category two had a very low number mixing ratio."*

3. Line 434: 'In this experiment' I suggest clarifying this as 'In experiment nCat2FFD' to avoid confusion since two experiments are stated in the first sentence.

*Thank you for the suggestion, it will be clarified.*

4. Line 437: 'a similar size to the original raindrops' And what size are they?

*The expression "the original raindrops" will be replaced with the expression "raindrops at the top of the subfreezing layer". We will also refer to Fig. B2 in the following sentence:*

*"The size distribution of the raindrops at the top of the subfreezing layer is presented by the dashed pink curve in Fig. B2."*

---

## Author Comment (AC4)

**Answers to the Community Referee Heather Reeves**

*Answers are in italic blue text.*

**CC1: 'Comment on egusphere-2024-594', Heather Reeves #3**

Recommendation: Accept with minor edits

Summary: This paper addresses advances to an NWP microphysics scheme that may allow for better prediction of ice pellets (PL). Specifically, it shows that secondary ice production (SIP) appears to have been pivotal for transitioning falling hydrometeors from all liquid to all ice, thus resulting in PL. Two processes that enhance the conversion from liquid to ice are parameterized in this paper (fragmentation of freezing drops FFD and Hallet-Mossep HM). Additional modifications were made to the FFD code to yield more representative results. This is a strong paper. It's clear and concise and the science is compelling. I have only minor thoughts below.

*Thank you.*

1.      Line 134: It says the two SIP processes are studied independently (as opposed to simultaneously including both FFD and HM in the same experiment) because that requires the default implementation of HM to be modified. I don't understand why HM would have to be modified. It is described as being a different physical process than FFD, so up to this point I thought these were 2 separate processes. Can the authors clarify?

*The sentences at lines 134-136 included too many details that were possibly confusing and misleading for the reader. The original sentence at lines 134-136 was:*

*"Observations collected in the field suggest that FFD and HM were active SIP processes during the 12 January 2020 ice pellet episode (LT22; Lachapelle et al., 2024).The two SIP processes were analyzed individually; no simulation used more than one SIP process simultaneously because this would result in the production of secondary ice through two distinct processes freezing the same raindrop and would require the default implementation of HM to be modified."*

*For clarity, the sentences will be changed to (changes in bold):*

*"Observations collected in the field suggest that FFD and HM were active SIP processes during the 12 January 2020 ice pellet episode (LT22; Lachapelle et al., 2024).* ***However, because the objective of this work is to examine how SIP affects simulated precipitation types and particle size distributions, the two SIP processes, HM and FFD, were used individually. This approach facilitated the understanding of their respective effects.****"*

*We will also add the following sentence mentioning future studies in the conclusion (near line 395):*

*"Future research should also include simulations combining multiple SIP processes, from which could emerge complex interactions and feedback processes."*

*In parallel, sensitivity tests have shown that adding HM to a simulation that includes FFD does not impact the accumulation of ice pellets much. In our parametrization of these processes, HM and FFD both occurs under similar conditions, when supercooled drops are collected by ice particles. Hence, because the parametrization of FFD is more efficient than the parametrization of HM, adding HM to FFD did not have much impact.*

2.      Line 138: The paper describes the number of ice splinters produced per unit of rain at a certain temperature range and how it changes outward from there. Is there a citation for this is or is this something the authors of this paper prescribed?  If the latter, how sensitive are the results to the number of ice splinters?

*The parametrization that we have used is the same as in Milbrandt & Morrison (2016). The numbers used in this parametrization come from Mossop et Hallet (1974). We will add these references in the sentence at line 138.*

*On another note, the simulations are sensitive to the parametrization of HM. In our answer to the second comment of referee #1, we changed the ice diameter threshold included in the parametrization of HM. Decreasing this threshold from 4000 µm to 1000 µm increased the amount of simulated ice pellets and decreased the amount of simulated freezing rain (Figs. R1, R2, R3).*

*Following these results, the following sentences will be added to the conclusion (near line 395):*

*"The simulations are sensitive to the parametrization of SIP. For example, increasing the ice diameter threshold for HM decreases the amount of ice pellets produced. The identification of an optimal parameterization of SIP for ice pellet and freezing rain simulation will require more observations and modeled cases."*

[Figure]

*Fig. R1. Ice pellet accumulation simulated with (a) nCat2_HM configuration that used an ice diameter threshold of 4000µm, (b) nCat2_HM1mm configuration that used an ice diameter threshold of 1000µm, and (g,h,i) their differences.*

[Figure]

*Fig. R2. Same as Fig. R1 but for freezing rain.*

[Figure]

*R3. Same as Fig. C1 in the submitted manuscript with the new simulation nCat2_HM1mm. In the revised manuscript, the simulation nCat2_HM will be replaced by nCat2_HM1mm.*

3.     I'm confused by the Appendices. It's not clear to me why they're included in the paper as appendices. I think Appendix A could be moved into Section 2. I struggled with Appendix B since its first referenced on line 155, before we know anything about the case study or experiments. I think the sensitivity tests in appendix B give some broader context to the results of this paper that merit putting this in the main body of the paper. And, like above, I think the content in Appendix C could just be put in the main part of the paper as well. Both Appendices B and C include good content, but having them as appendices confuses me as a reader. Moving that content into the main body of the paper will strengthen the story line and give greater import to the creative work presented in these parts of the paper.

*Thank you for the suggestion. We agree that the appendices could be moved into the main body of the text. However, although they present valuable results, we decided to leave them as appendices because they allow the manuscript to keep a better focus and flow between the Methodology and Results sections.*

4.      Paragraph starting at line 220: I like the air parcel trajectory approach, but I'd like to know to what degree that trajectory bobs up and down in the vertical. I think a simple way to address this is to add an inset to Fig. 1 that shows a vertical cross section along the trajectory that shows the position of the parcels from each experiment as a function of time/location. That way the reader can assess whether the changes made to the microphysics scheme impact the rate at which the parcels are advected and whether their vertical ascent/descent differs.

*Fig. R4 shows the trajectory vertical heights for each experiment (Fig. R4).*

[Figure]

[Figure]

*Fig. R4. Simulated trajectories height.*

*We will add the trajectory vertical height (Fig. R4) in Fig.1 (Fig. R5). Only nCat1_noSIP will be shown in Fig. 1. We think that including additional curves to the figure will bring confusion.*

[Figure]

*Fig. R5: Revised Fig. 1. The panel in the lower left shows the vertical height of the trajectory for experiment nCat2_noSIP; the trajectories calculated with experiments nCat2_HM, nCat2_FFD, and nCat2_FFD_MOD reached similar results (not shown).*

5.    I find Figs. 3,4,5 difficult to read. It's a lot of skinny lines and some colors are difficult to distinguish and some of the lines overlay each other enough to make it hard for the reader. I wonder if the authors would consider converting this to a "chicklet plot" for the ptype forecasts. This would be a lot easier for the reader to interpret.  It would require putting the rates in a separate panel, but I think it's worth the extra real estate to make a clearer graphic.

*We have improved these three plots by making them more compact (Figs. R6-R8 which will become revised Figs. 4-6). Observed precipitation rates for Ottawa will also be added (black dashed line in Fig. R8).*

[Figure]

*Fig. R6: Revised Fig. 3. (a) Hourly simulated and reported precipitation types at UQAM-PK for nCat1_noSIP, nCat2_HM1mm, nCat2_HM1mm_FFD, and nCat2_HM1mm_FFD_MOD. Precipitation types are rain (green), snow (blue), freezing rain (red), and ice pellets (purple). Note that between 0430 and 1600 UTC 12 January 2020, the macro photography analysis revealed the presence of tiny ice crystals (~ 200 μm) mixed with ice pellets. These were too small to be reported by manual observers. (b) Total precipitation rate simulated and observed at UQAM-PK for the same simulations. The dashed black line shows the precipitation rate measured by a single-Alter Geonor.*

[Figure]

*Fig. R7: Revised Fig. 4. Same as Fig. R6 but at Mirabel International Airport. The manual observations were conducted hourly. The measured precipitation rate was not available for this location.*

[Figure]

*Fig. R8: Revised Fig. 5. Same as Fig. 3 but at Ottawa International Airport. The manual observations were conducted hourly. The precipitation rate was measured by a rain gauge installed at Ottawa airport.*

6.      It's interesting to me that in Fig. 6 the precip rate varies between the experiments (this is also evident in Figs. 3-5). Can the authors add some thoughts to the paper on why this is?

*The following sentences will be added after the second paragraph in section 4.1:*

*"Adding SIP and other modifications had a non-negligible impact on the simulated precipitation rate because it impacted the particle size distribution and fall velocity. Smaller simulated particles fall at a slower velocity and are advected over longer distances by horizontal wind. In contrast, larger and denser ice particles fall at a higher velocity and reach the surface closer to their point of origin (e.g. Thériault et al. 2012[1]). This behavior suggests that simulating the accurate size distributions would improve the simulated precipitation rate. In section 4.2, we show that the hydrometeor size distributions simulated by nCat2_FFD_MOD were similar to those observed, unlike in the other experiments."*
* * *
[1] Julie M. Thériault, Ronald E. Stewart, et William Henson, « Impacts of terminal velocity on the trajectory of winter precipitation types », *Atmospheric Research*, Remote Sensing of Clouds and Aerosols: Techniques and Applications - Atmospheric Research, 116 (15 octobre 2012): 116-29, https://doi.org/10.1016/j.atmosres.2012.03.008.

7.    I see there's MMR data that shows the vertical level at which the transition from FZRA to PL occurs (line 291). Out of curiosity, have the authors tried to reproduce synthetic MMR data from the simulation to see if the transition from FZRA to PL in the vertical is accurately handled by the FFD experiments?

*Radar reflectivity diagnosed by P3 can be compared with remote sensing observations. Figs. R9 and R10 show the vertical reflectivity measured by the MRR2 installed at UQAM-PK in downtown Montreal, and the quasi-vertical reflectivity measured by Environment and Climate Change Canada scanning S-band radar in Blainville (near Mirabel), compared to the radar reflectivity simulated by nCat2_FFD_MOD.*

*The quality of the comparisons could be improved if the reflectivity was outputted at a higher temporal frequency. Overall, we see good agreement between the observations and the simulation at both locations, especially at the top of the melting layer. Additionally, the decrease in simulated reflectivity below 500 m between 0800 and 1400 UTC 12 January 2020, which was associated with ice pellet formations, compared well with the observed reflectivity at the CASBV radar (Fig. R10).*

[Figure]

*Fig. R9. Comparison between vertical reflectivity measured by UQAM-PK MRR-2 vertically pointing radar and simulated reflectivity extracted from the three-dimensional model at UQAM-PK coordinates.*

[Figure]

*Fig. R10. Comparison between quasi-vertical reflectivity measured by CASBV ECCC scanning radar (near Mirabel) and simulated reflectivity extracted from the three-dimensional model at CASBV coordinates.*